# Characteristics of Environmental *Klebsiella pneumoniae* and *Klebsiella oxytoca* Bacteriophages and Their Therapeutic Applications

**DOI:** 10.3390/pharmaceutics15020434

**Published:** 2023-01-28

**Authors:** Beata Weber-Dąbrowska, Maciej Żaczek, Małgorzata Łobocka, Marzanna Łusiak-Szelachowska, Barbara Owczarek, Filip Orwat, Norbert Łodej, Aneta Skaradzińska, Łukasz Łaczmański, Dariusz Martynowski, Marta Kaszowska, Andrzej Górski

**Affiliations:** 1Bacteriophage Laboratory, Ludwik Hirszfeld Institute of Immunology and Experimental Therapy, Polish Academy of Sciences, 53-114 Wrocław, Poland; 2Phage Therapy Unit, Ludwik Hirszfeld Institute of Immunology and Experimental Therapy, Polish Academy of Sciences, 53-114 Wrocław, Poland; 3Institute of Biochemistry and Biophysics, Polish Academy of Sciences, 02-106 Warsaw, Poland; 4Department of Biotechnology and Food Microbiology, Faculty of Biotechnology and Food Science, Wrocław University of Environmental and Life Sciences, 51-630 Wrocław, Poland; 5Laboratory of Genomics & Bioinformatics, Ludwik Hirszfeld Institute of Immunology and Experimental Therapy, Polish Academy of Sciences, 53-114 Wrocław, Poland; 6Laboratory of Microbial Immunochemistry and Vaccines, Ludwik Hirszfeld Institute of Immunology and Experimental Therapy, Polish Academy of Sciences, 53-114 Wrocław, Poland; 7Infant Jesus Hospital, Medical University of Warsaw, 02-005 Warsaw, Poland

**Keywords:** antibiotic resistance, hypervirulent strains, multi-drug resistant strains, ESBL, *Klebsiella*, phage biology, phage therapy, emerging pathogens, *Slopekvirus*, *Jiaodavirus*

## Abstract

In recent years, multidrug-resistant (MDR) strains of *Klebsiella pneumoniae* have spread globally, being responsible for the occurrence and severity of nosocomial infections. The NDM-1-kp, VIM-1 carbapenemase-producing isolates as well as extended-spectrum beta lactamase-producing (ESBL) isolates along with *Klebsiella oxytoca* strains have become emerging pathogens. Due to the growing problem of antibiotic resistance, bacteriophage therapy may be a potential alternative to combat such multidrug-resistant *Klebsiella* strains. Here, we present the results of a long-term study on the isolation and biology of bacteriophages active against *K. pneumoniae*, as well as *K. oxytoca* strains. We evaluated biological properties, morphology, host specificity, lytic spectrum and sensitivity of these phages to chemical agents along with their life cycle parameters such as adsorption, latent period, and burst size. Phages designated by us, vB_KpnM-52N (Kpn52N) and VB_KpnM-53N (Kpn53N), demonstrated relatively broad lytic spectra among tested *Klebsiella* strains, high burst size, adsorption rates and stability, which makes them promising candidates for therapeutic purposes. We also examined selected *Klebsiella* phages from our historical collection. Notably, one phage isolated nearly 60 years ago was successfully used in purulent cerebrospinal meningitis in a new-born and has maintained lytic activity to this day. Genomic sequences of selected phages were determined and analyzed. The phages of the sequenced genomes belong to the *Slopekvirus* and *Jiaodavirus* genus, a group of phages related to T4 at the family level. They share several features of T4 making them suitable for antibacterial therapies: the obligatorily lytic lifestyle, a lack of homologs of known virulence or antibiotic resistance genes, and a battery of enzymes degrading host DNA at infection.

## 1. Introduction

According to the global priority list of antibiotic-resistant bacteria published by the World Health Organization in 2017, *Klebsiella* strains have been listed among the deadliest bacteria that pose an exceptional treat to human health, especially in hospitals and nursing homes [1]. This list is intended to spur governments to prioritize research and development toward new antimicrobials. The panel of WHO experts emphasize that the current data might be underestimated due to a lack of high-quality data from low-income countries and surveillance data on livestock and food.

*Klebsiella* tends to colonize the mucosal surface in humans making those strains responsible for a plethora of infections, i.e., respiratory tract infections, urinary tract infections (UTIs), bloodstream infections, and liver abscesses [2]. In recent years *Klebsiella* has been considered the main factor causing nosocomial infections with an increasing mortality rate [3]. Notably, *Klebsiella* is capable of causing primary pneumonia, a feature which is rare among other Gram-negative rods [4].

The importance of actions that need to be taken in order to reduce the spread of antibiotic-resistant bacteria has been illustrated in a growing number of published papers. The authors emphasize that *K. pneumoniae’s* large accessory genome is not only responsible for their hypervirulence and multidrug resistance but also help them to successfully colonize clinical and environmental niches [4,5]. For example, Hernandez-Garcia et al. [6] described an outbreak caused by NDM-1+CTX-M-15+DHA-1-producing *K. pneumoniae* (NDM-1-Kp) in Spain. A total of 9 patients hospitalized in the same health care facility were diagnosed with NDM-1 (New Delhi metallo-ß-lactamase 1)-positive strains. Based on the obtained results, the authors emphasize rapid disseminations of NDM-1 carbapenemase among strains with strong persistence in the hospital setting. The relationship between the hospital environments and the prevalence of antibiotic resistance in *Klebsiella* strains was confirmed much earlier by Struve and Krogfelt [7] who did not find any antibiotic resistant strains (except ampicillin) among environmental isolates. Notably, these strains turned out to be as virulent as their clinical origins which, in the case of acquiring antibiotic resistance genes, could make them pathogens that are difficult to combat. A decade later, Struve et al. [8] using the whole-genome sequencing revealed that a clonal complex 23 (CC23) grouping hypervirulent *Klebsiella* isolates had already spread globally and that the process was rapid. One explanation might be the diversity found in the iron acquisition systems detected in CC23. Notably, the pathogenicity of *Klebsiella* stains is driven by the ability to acquire iron. The widespread presence of *Klebsiella* carbapenemases was also noted by Yoon et al. [9]. While in 2013, only one carbapenemase-positive strain was isolated from Korean hospitals, this number rose to 348 in 2015 (among which the presence of the *bla*KPC gene responsible for rapid dissemination was confirmed). Analogously, among 137 patients investigated at eight New York and New Jersey medical centers in the years 2016–2021, 65% of them had KPC-producing bacteria [10]. Swathi et al. [11] concluded that there is an urgent need to further explore the *bla*KPC gene and to implement appropriate detection methods as its unstable region may be involved in variations of *K. pneumoniae* carbapenemases. It has been proven that patients dealing with infections caused by carbapenem-resistant strains tend to wait longer until appropriate therapy is applied and experience higher mortality rates [10].

Numerous microbiome analyses revealed that hypervirulent *Klebsiella* strains are more common in stool samples collected from hospitalized patients compared to healthy individuals which likely contributes to their widespread distribution in health care facilities. There is a belief that gut colonization by hypervirulent *Klebsiella* triggers subsequent infection in the same patient [4]. Kamau et al. [12] highlight that most patients dealing with hypervirulent *Klebsiella* infections denied any recent international travel suggesting that such strains are no longer endemic to their original habitats (i.e., Asian countries).

Of note, *K. pneumoniae’s* complex consists of seven *K. pneumoniae*-related species, including newly described *Klebsiella variicola* [13]. The authors report that this strain is responsible for a higher, compared to *K. pneumoniae*, mortality rate in patients suffering from bloodstream infections. Both strains can be characterized by a hypermucoviscous-like phenotype that potentially triggers virulence. The increasing number of such strains has gained much attention among clinicians [14]. It is pertinent to mention that another *Klebsiella* species, *K. oxytoca*, is becoming increasingly associated with nosocomial infections, particularly in immunocompromised patients. They are also causative agents of pediatric antibiotic-associated hemorrhagic colitis [3].

In view of the above, the aim of this paper is to bring closer attention to phages against *Klebsiella* strains with their potential use in therapy. Given the fact that phages are the most abundant entities on the planet and the number of phage particles circulating in the environment, including the human body, exceeds 10 times the number of bacteria, their mutual interactions seem to be inevitable and can have an impact on clinical outcomes of such treatment [15,16,17]. Of note, the complexity of phage-host interactions remains largely unexplored [16]. Moreover, phage growth and cultivation in vitro does not necessarily mean that similar properties will be maintained in vivo [18]. Nevertheless, attempts at successful phage therapy against *Klebsiella* infections have been widely described in the literature [19,20,21,22,23,24,25,26] including data obtained in Phage Therapy Unit (PTU) at the Hirszfeld Institute of Immunology and Experimental Therapy (HIIET) in Wrocław, Poland [27,28,29,30]. Equally numerous are articles focusing on the isolation of *Klebsiella* phages and their biology [3,31,32,33]. Interestingly, one phage isolated by Przondo-Hessek [34] in the 1960s, named vB_KpnM-9, was successfully used in purulent cerebrospinal meningitis in a new-born [35] and still displays lytic activity against pathogenic *Klebsiella* strains. Additionally, a newly isolated phage from our collection was used in a 60-year-old kidney transplant recipient with recurrent urinary tract infections [30].

## 2. Materials and Methods

### 2.1. Bacterial Strains and Growth Conditions

A total of 29 *Klebsiella* strains were isolated from environmental samples and city sewage deposited in the Bacteriophage Laboratory (BL) of HIIET in the years 2005–2015. In addition, in the years 1976–2021 we received 749 clinical strains of *K. pneumoniae, K. oxytoca*, *K. variicola*, *K. ozenae* and *K. aerogenes* from hospitals and teaching departments located throughout the entire country, including PTU at the HIIET, “Diagnostyka”, “Practimed”, “Dialab” diagnostic laboratories, the 4th Military Hospital in Wrocław, the University Teaching Hospital in Opole and the Department of Medical Microbiology of the Warsaw Medical University. We also used *Escherichia coli* B and *K. pneumoniae* ATCC 13886 from the Polish Collection of Microorganisms at HIIET as reference strains.

Bacterial strains were cultured routinely under aerobic conditions on liquid media: nutrient broth (BioMaxima S.A., Lublin, Poland) and Luria-Bertani (LB) broth (Sigma-Aldrich, Merck, Poznań, Poland) as well as solid media. The following agar plates (BioMaxima S.A., Lublin, Poland) were used: MacConkey as a selective and differential medium, Mueller-Hinton to assess host range and lytic spectrum and NZCYM LAB-AGAR was used for cultivation of the *E. coli* strain. All strains were incubated at 37 °C for 18 h with shaking (150 rpm). In order to isolate single colonies on agar plates, a streaking method was utilized.

The majority of strain identification was carried out on a Vitek 2 Compact analyzer (bioMerieux), an automated microbial identification system. We used the manufacturer’s GN cards (for Gram-negative bacilli) and GP cards (for Gram-positive cocci). At this point it is worth mentioning that rapidly changing bacterial taxonomy and the emergence of newer methods of identification will certainly change the taxonomy in the future. Thus, we cannot be absolutely convinced that the strains described in this paper will not turn out to be different *Klebsiella* species or even different bacterial genera. The phenotypic and biochemical characteristics of the *K*. *pneumoniae* complex are difficult to distinguish, especially using standard biochemical techniques [13]. Thus, inconsistencies in bacterial identification constitute an ongoing issue. De Campos et al. [36] recently described five *K. variicola* strains initially identified as *K. pneumoniae*, with a Vitek-2 System and 16S rRNA sequencing. However, a one-step multiplex polymerase chain reaction and Whole Genome Sequencing (WGS) identified them as *K. variicola*. Similarly, some of the *K. pneumoniae* strains deposited in our laboratory turned out to be *K. variicola.* By analogy, using the MALDI-TOF mass spectrometry system [37] we found that one bacterial strain from our collection, previously classified as *Klebsiella*, currently belongs to the *Raoultella* genus [38] and represents *Raoultella planticola*. Hence, to reflect the change in classification we have added a new symbol (Rpl1) for the phage previously designated as Kpn2. Certain other bacterial species, formerly named *Klebsiella*, are classified as *Enterobacter, Pantoea* or *Pluralibacter* (Table 1).

As mentioned above, for more precise identification, the MALDI-TOF MS Biotyper platform was used to identification of bacteria and analyses of affinity of all strains. Intact bacterial cells were deposited on a target plate with α-cyano-4-hydroxy-cinnamic acid in a mixture composed of acetonitrile (50%), water (47.5%) and trifluoroacetic acid (2.5%) as a matrix (10 mg/mL) and analyzed using the Ultraflextreme instrument (Bruker, Germany) with Biotyper software (version 3.0) supplemented with a database of 8468 reference strains. The raw spectra were recorded in the positive linear mode within a mass range of 2–20 kDa.

### 2.2. Bacteriophage Origin

A total of 101 *Klebsiella* bacteriophages were used in the study, including 35 bacteriophages obtained from the Przondo-Hessek collection in 1974 as a result of scientific cooperation [34,39]. 33 bacteriophages were isolated from city sewage, whereas 66 phages were isolated from environmental samples used in the BL of HIIET for routine phage isolation (Table 2). Two phages were isolated from biological samples collected from patients (nose swab and stool sample). Phages were isolated with the use of 121 clinical strains selected from 749 strains deposited in our laboratory.

### 2.3. Phage Isolation from Environmental Samples

For phage isolation we screened 158 environmental samples deposited at our laboratory. These samples originated from numerous water reservoirs located in Poland (sea water, inland water and tap water) along with city, hospital, and industrial sewage samples (the latter ones were collected in the years 2000–2016 from various water treatment plants in Poland). Raw samples as well as those incubated in peptone water were used. After incubation samples were filtered through membrane filters with a pore diameter of 0.22 μm (Millipore, Millex-GR, PES membrane, Burlington, MA, USA). Some of the samples were concentrated by cellulose diacetate dialyzers to a volume of 50–100 mL to increase the probability of phage presence (membranes with nominal molecular weight limit of 1 kDa were used). To accelerate sample screening, we also used tetrazolium as a measure of bacterial cell growth after sample treatment with concentrated filtrates. Application of tetrazolium is based on enzymatic reduction of 2,3,5-triphenyltetrazolium chloride to red colored triphenylformazan by live bacteria cells. The final quantity of formazan (measured spectrophotometrically at 595 nm) corresponds with the number of viable bacteria cells and with phage particles in the tested sample. Tetrazolium was used according to McLaughlin [40] with some changes. Briefly, 96-well microplates (Sarstedt) were loaded with 100 µL of bacterial culture (10^6^ cfu/mL) and 100 µL of the appropriate environmental sample per well. Bacterial dilutions were prepared in sterile peptone water. Such loaded microplates were incubated under two different conditions (for 6 h at 37 °C and for 18 h at 25 °C). The last component, tetrazolium suspension (10 mg/mL of peptone water), was added in the dark to the final concentration of 2 mg/mL at the final stage of incubation (tetrazolium is light sensitive). All stages were prepared aseptically in a laminar flow cabinet. Positive results indicating bacteriophage presence were always confirmed on Mueller-Hinton agar plates by spotting the filtered content of a specific well onto the bacterial lawn grown from the strain used in the experiment.

### 2.4. Phage Propagation and Titer Determination

The bacterial culture was prepared in 10 mL of nutrient broth prepared with milliQ water by inoculating a single colony of a strain taken from a solid medium. The bacterial culture was incubated at 37 °C with shaking at 150 rpm for 18 h. Further stages of phage propagation were performed according to Żaczek et al. [41] with further modifications. The obtained lysate was filtered through membrane filters with pore diameters of 0.45 µm and 0.22 µm (Millipore Millex-GR, PES membrane, Burlington, MA, USA) and its titers were determined using the double agar layer method according to Adams [42] and Kropinski et al. [43] with some modifications. The initial step was to prepare bacterial suspension by adding an isolated single colony of the bacterial host to 10 mL of LB broth and incubating it at 37 °C for 3 h to achieve 1.5–2 of McFarland standard turbidity. Next, 50 µL of tested phage lysate was added to 450 µL of LB broth and used to prepare serial dilutions (from 10^−1^ do 10^−8^). In the next step, 100 µL of bacterial suspension was mixed with 200 µL of the prepared host strain and with 3 mL of 0.7% agarose solution warmed to approx. 40 °C. The newly created suspension was immediately poured onto the surface of Mueller-Hinton agar plates (in triplicate). Plates were incubated at 37 °C for 2–3 h and then at 4 °C overnight. Following incubation, plaques were visualized as zones of clearing in the bacterial lawn and could easily be counted.

In addition, the routine dilution test (RTD) was used to determine the lowest phage dilution that still causes clear confluent lysis of the bacterial host by spotting phage dilutions described above directly onto the bacterial lawn (with the omission of 0.7% agarose solution).

### 2.5. The Influence of Physical and Chemical Factors on the Lytic Activity of Phages

The lytic activity of 10 *Klebsiella* phages was examined under various storage conditions and under the influence of physico-chemical factors. We chose to test phages with the broadest lytic spectra that are considered potential candidates for phage therapy. Phage stability was assayed at 4 °C, at room temperature (approx. 25 °C), at −70 °C and in 25% glycerol at −70 °C after 3 and 12 months. Additionally, the stability of phage lysates stored at 4 °C was evaluated after 24 months. We also examined the effect of high temperature (60 °C) on the phage lytic activity for a period of 10 min as well as the influence of various chemical factors such as chloroform and pH between 4 and 8. The reference phage was suspended in nutrient broth (pH 7.1) (5.4 g of enzymatic hydrolysate of casein, 4 g of Bacto Pepton, 3.5 g of NaCl, 1.7 g of yeast extract, 0.4 g of beef extract and 1000 mL of miliQ water). Following incubation times, phage activity was examined with the double agar layer method described above using agar plates whose composition was developed especially for the cultivation of phages against Gram-negative bacteria (NaCl, Na_2_HPO_4_, enzymatic hydrolysate of casein, beef extract and agar). The effect of chloroform was evaluated for 2 h and 24 h. Phage activity at various ranges of pH (4, 5, 6 and 8) was tested after 1 h and 5 h at room temperature and then compared to the control lysate incubated at pH 7.1. The above-mentioned experiments were carried out according to the previously described methods with further modifications [44,45].

### 2.6. Host Range Determination

For host range determination, we used strains classified as *K. pneumoniae*, *K. oxytoca*, *K. variicola*, *K. aerogenes*, *Enterobacter cowanii, P. gergoviae*, and *P. agglomerans*. The strains used in this study are listed in Table 1. Overall, 462 strains in total were tested against 91 phages from our collection according to the method described by Weber-Dąbrowska et al. [18] and Kęsik-Szeloch et al. [46] with some amendments. Bacterial strains were stored at −70 °C in LB broth (Sigma-Aldrich, Merck, Poznań, Poland) with an addition of 25% glycerol. Prior to testing, frozen aliquots were resuspended in LB broth, incubated for 4–6 h at 37 °C and these prepared liquid cultures were poured into Mueller-Hinton agar plates (bioMerieux, Poland). After drying, a drop of the tested phage was spotted onto the bacterial lawn. Usually, the agar plate was divided into seven individual sectors so that seven different phages could be tested on one agar plate with a proper distance between spots maintained to minimize adjacent spots running together. Plates were incubated at 37 °C for 3–4 h and then at 4 °C overnight. Following incubation, plaques were visualized as zones of clearing in the bacterial lawn and could easily be counted.

### 2.7. One-Step Growth Experiments

The one-step growth curve is a classic study where the increase in infectious virus over time is followed by sequential sampling and titration. Shortly after adsorption and successful infection, the inoculated cell-free virus disappears (the so-called latent period). Following calculation of adsorption rate, we can calculate both a latent period and burst size. For lytic phages, latent-period termination occurs at lysis (phage-progeny release). The average number of phages released from a single bacterial cell is called the burst size. The optimal latent period is a period that maximizes the growth of a specific phage population under given conditions (i.e., quantity and quality of host cells) [47]. In general, bacteria more rapidly infected by phage particles and a higher burst size mean a more efficient and shorter phage life cycle that leads to the destruction of a specific bacterial population. To determine one-step growth components, we used the methodology described by Hyman and Abedon [48] with some modifications. Briefly, the density of 40 mL of overnight bacterial culture was measured at OD 600 nm and adjusted in peptone water to 10^6^ CFU/mL. The phage lysate was also diluted to achieve a titer of 10^6^ PFU/mL. Next, 10 mL of bacteria suspension was mixed with 10 mL of phage lysate in order to achieve a MOI = 1. The mixed culture was incubated at 37 °C for an 80–90 min period, during which every 10 min a sample of culture was taken and filtered through membranes (0.22 µm, Millipore Millex-GR, PES membrane, Burlington, MA, USA). Subsequently, serial dilutions of the collected samples were immediately prepared and free phage particles were enumerated using the double agar layer method described above. Phage adsorption was expressed as the highest percentage of reduction of the titer in the supernatant compared to the control (time 0 sample) titer. The latent period was defined as the time between bacteriophage adsorption and the first release of newly formed phage particles, whereas burst size was calculated as the ratio of the titer of released phage particles obtained within the duration of two latent periods to the number of infected bacterial cells during bacteriophage adsorption. One-step growth testing was performed on the selected phages thanks to their broadest lytic spectra.

### 2.8. Electron Microscope Study

#### 2.8.1. Phage Preparation

Phage preparation for electron microscopy was carried out according to a method described by Ackermann [49] and in our previous article [46]. Briefly, a sterile, high-titer (10^9^–10^10^ pfu/mL) lysate was centrifuged for 60 min at 25,000× *g*. The supernatant was discarded, replaced by a smaller volume of 0.1 M ammonium acetate and gently mixed by pipetting. A drop of washed phage suspension was deposited on formvar/carbon supported copper grids and left to absorb for 1 min. Then, the liquid was drained off with filter paper. After that, a drop of 2% uranyl acetate was placed on a grid and drained after 1 min. This stained preparation was examined in the transmission electron microscope (TEM). The transmission electron microscope Jeol JEM- F 200 with accelerating voltage 80 KV and TVIPS TemCam camera (XF-Series) were used for phage morphology studies.

#### 2.8.2. Phage Particles Measurements

Phage particles were measured on images obtained from TEM microscopy using ImageJ software. At least 10 well-contrasted virions were used to establish each dimension. We sought for angular capsids with parallel sides. All positively stained phages were rejected.

### 2.9. Plaque Morphology Analysis

Plaque morphology was performed on Mueller-Hinton agar plates. We focused on plaque turbidity and diameter (both external and internal for plaques with halos). The method was carried out using the double agar layer assay described above.

### 2.10. Phage DNA Isolation, Phage Genome Sequencing and Bioinformatic Analysis

Total phage DNA was isolated from the phage lysate. Briefly, 2 mL of lysate (1 × 10^9^ pfu/mL) was concentrated 10× with an Amicon (Millipore, Amicon Ultra—0.5 mL of UFC501024 or 2 mL of UFC201024). DNA was isolated from concentrated lysates using QIAamp MinElute Virus Spin Kit (Qiagen, Hilden, Germany) according to the manufacturer’s protocol. DNA concentration was measured with the fluorometric method (Promega, Madison, USA). Library preparation was performed using an Ilumina DNA Prep kit (cat. 20018705) according to the Reference Guide (Illumina, San Diego, CA, USA). Library quality control was performed using the High Sensitivity D1000 ScreenTape System on TapeStation (Agilent Technologies, Waldbronn, Germany). Fluorometric DNA quantitation was carried out with Quantus, Promega. The sample was diluted according to MiSeq System Denature and Dilute Libraries Guide (Protocol A) (Illumina, San Diego, CA, USA), as well as pooled and sequenced on MiSeq (Illumina, San Diego, CA, USA) using MiSeq Regent Kits v2, 300-cycles. The quality of the sequence reads was verified using FastQC, https://www.bioinformatics.babraham.ac.uk/projects/fastqc/ (accessed on 19 December 2022). De novo assembly was performed using unicycler [50] and spades v3.14.0 [51]. For each sample, a contig corresponding to the entire phage genome was generated.

Each assembly was additionally verified with the use of a de novo sequence assembler of Geneious Prime software version 2021.2.1 (Biomatters, Auckland, New Zealand). The determination of phage packing strategy was performed based on the results of PhageTerm [52] analysis using all sequence reads of a given phage as the input file, and on the results of manual inspection of the sequence reads assembly pattern. Putative protein coding genes and genes encoding tRNAs were predicted with the use of RAST server [53,54] and tRNAscan-SE [55], and manually corrected. Annotations of each genome were supplemented based on a comparison with the annotations of related genomes in GenBank (accessed on 28 July 2021) with the use of Blastp and tBlastn [56,57,58]. A search for unique sequences in the genome of each phage was performed with the use of Panseq [59]. The numbers of homologous proteins between phages were determined with the help of CoreGenes5 [60,61] with parameters set for the bidirectional best hit and E values set for more stringent (E = 10^−5^) or less stringent (E = 0.01). A search for putative genes encoding antimicrobials was performed with the use of ResFinder and an updated database of resistance determinants to 119 antimicrobials of 22 different classes [62] (https://cge.cbs.dtu.dk/services/ResFinder/). The prediction of putative homologs of known virulence factors was performed using the on-line version of the similarity-based classifier of VirulentPred and a RefSeq database of proteins http://203.92.44.117/virulent/submit.html (accessed on 19 December 2022) [63] and further verified by inspecting the structural homologs of each positive hit based on the results of HHPred [64]. Additionally, this was supported by the manual inspection of functional assignments of predicted gene products or their homologs.

### 2.11. GenBank Accession Numbers

Complete genomic sequences of phages Kpn35c1, Rpl1, Kpn5N and Kpn6N have been deposited in GenBank, under the accession numbers: OK631813, OK631813, MZ890186 and OK631813, respectively.

## 3. Results and Discussion

### 3.1. Bacteriophage Presence in Environmental and Sewage Samples

The presence of *Klebsiella* phages was observed in 20 out of 158 water samples as well as samples collected from water treatment plants and hospital sewage (Table 2). For phage procurement, 121 *K. pneumoniae* and *K. oxytoca* (79 and 42 respectively) strains were used including ESBL-producing clinical strains. Among these 121 strains, 50 of them, namely 34 *K. pneumoniae* and 16 *K. oxytoca* strains, were selected as host strains for 65 newly isolated bacteriophages (36 for *K. pneumoniae* and 29 for *K. oxytoca* respectively).

Samples collected from the Vistula River and city sewage from a wastewater treatment plant in Opole turned out to be the samples with the most abundant presence of *K. pneumoniae* isolates (16 and 12 phages respectively). Contrary to expectations, we were able to isolate new phages only from inland waters and sewage samples. This may be associated with the use of clinical strains as preys in phage isolation. Such strains are also abundant in sewage samples deposited in our sample collection, and most of our phages were isolated from sewage.

The *K. oxytoca* phages were most frequently found in the city sewage from wastewater treatment plants in Kraków-Płaszów (6 phages) and in the incubated city sewage in Opole (5 phages) as well as in a crude water sample taken from a water treatment plant in Grodzisk Mazowiecki (5 phages).

### 3.2. Bacteriophage Host Range Determination

As previously mentioned, a total of 462 strains classified as *K. pneumoniae*, *K. oxytoca*, *K. variicola*, *K. aerogenes*, *K. ozenae*, *R. planticola*, *P. agglomerans*, *P. gergoviae*, and *E. cowanii* (Table 1) were tested against 91 *Klebsiella* phages.

Among all tested phages, no lytic activity was observed against *K. ozenae*, *E. cloacae* and *E. coli* B strains.

Overall, 749 *Klebsiella* clinical strains (592 *K. pneumoniae*, 144 *K. oxytoca*, 2 *K. aerogenes*, 9 *K. ozenae* and 2 *K. variicola*), including the aforementioned 462 strains used in host range experiments, isolated from patients with different categories of the diseases, collected in the years 1978–2021 have been tested with *Klebsiella* phages from the Przondo-Hessek historical collection and HIIET phage collection (the isolation of *Klebsiella* phages has been performed since 2009). For both collections, phage lytic activity was observed in 382 (51.1%) strains, including 315 (82.4%) for *K. pneumoniae*, 65 (17.1%) for *K. oxytoca* and 2 (0.5%) for *K. variicola*. No lytic activity was found for 365 (48.9%) strains, including 277 (75.9%) for *K. pneumoniae*, 79 (21.6%) for *K. oxytoca* and 9 (2.5%) for *K. ozenae* strains.

A comparative analysis of the lytic spectrum carried out between these two phage collections in the years 2009–2021 with the use of 207 clinical *K. pneumoniae* strains demonstrated a rather narrow range of phage activity (Table 3) with some exceptions (rates above 30%). These results are in line with data described in the literature [44].

The highest lytic spectrum was observed for the lytic *K. pneumoniae* 53N and 5N phages (35.2% and 28.5% respectively) isolated from incubated hospital sewage from the 4th Military Hospital in Wrocław (53N) and from incubated city sewage from a wastewater treatment plant in Kraków-Płaszów (5N). It is worth noting that *Klebsiella* phages from the Przondo-Hessek collection isolated in the 1960s show similar lytic activity against the currently isolated clinical *K. pneumoniae* strains.

In the same period covering the years 2009–2021 we investigated the host range of *K. oxytoca* phages using 106 clinical strains identified as *K. pneumoniae* (86), *K. ozenae* (9), *K. aerogenes* (8) and *K. variicola* (3). It was found that among 86 *K. pneumoniae* strains 38 (44.1%) were sensitive to *K. oxytoca* phages, the newly isolated phage with symbol Klox64N, and phage *K. oxytoca* Klox3 (from the Przondo-Hessek historical collection) demonstrated the broadest lytic spectra, i.e., 40% and 28% respectively. No lytic activity was observed among *K. ozenae* strains.

Quite different results were obtained with the use of 160 *Klebsiella* strains (103 from the HIIET collection and 57 obtained from the Department of Medical Microbiology at the Medical University of Warsaw). These clinical strains were collected in the years 2006–2021 and they were isolated from patients with urinary tract infections. More than 78% of strains (including ESBL-positive ones) were sensitive to phages from the HIIET collection. Three phages from our collection, *K. pneumoniae* 52N, 53N and *K. oxytoca* 56N, isolated from different environmental samples, demonstrated the broadest lytic spectra: 55.1%, 48.3% and 44.9% respectively.

Taken together, only some tested phages showed relatively broad lytic spectra (approx. 35–50%, dependent on the strain collection used) compared to our other isolates. The majority of phages could infect only between 9–30% of the tested strains. Such a correlation has been observed in numerous phage-host systems where extensive prevalence of a given bacterial host is associated with the predominance of narrow host range phages, while high diversity in hosts promotes the propagation of broad host range phages [16]. A similar narrow host range for *Klebsiella* phages, including those lytic against a multiresistant or KPC-producing strain, was described by other authors [31,65].

### 3.3. One-Step Growth Experiments

The results from one-step growth testing are summarized in Figure 1a–d. Overall, the adsorption rate ranged from 28.5% to 61% with a median of 47.7%, adsorption time ranged from 10 to 30 min with a median of 20 min, the latent period varied between 20 and 40 min (a median of 30 min). The greatest variability was noticed for burst size which varied from 5 to 75 pfu/cell with a median of 32 pfu/cell. All experiments were performed in triplicate and the provided results constitute the mean value of each set of repetitions.

All tested phages indicated a rapid increase in phage titer within 80 min after infection at initial MOI = 1 (Figure 2a–c). Interestingly, adsorption of the tested phages was much lower when compared to phages described in the literature [45,46] and its peak was noted later (10 to 30 min.) (Figure 1a). The differences could result from different experiment conditions (i.e., different MOI used by the authors). Nevertheless, phage infection kinetics presented in Figure 2a–c did not show any deviations. To increase readability, we gathered all tested phages in three groups (each group per one figure).

Most of the tested phages formed clear plaques suggesting their lytic life cycle. In three cases plaques appeared turbid, one phage produced clear plaques with a halo. However, it must be emphasized that these occurrences could be caused by external factors and not phage characteristics [44]. Summarized data regarding plaque morphology are presented in Figure 3.

### 3.4. Bacteriophage Morphology Using Electron Microscopy

Phages from our new collection (phages isolated since 2009) were classified to the tailed phages of the myovirus, siphovirus of podovirus morphotype. *Klebsiella* phages of similar morphotypes have been also described by other authors [17,33,46,66]. The most numerous in our collection are phages of the myovirus and podovirus morphotype (26 phages each), whereas only 4 phages belong to the siphovirus morphotype. The morphotype C1 was found in the case of 26 phages (15 *K. pneumoniae*, 11 *K*. *oxytoca*) whereas morphotype A2 was found in the case of 16 phages (14 *K. pneumoniae*, 2 K. *oxytoca*). Phages with the broadest lytic activity, except for phage vB_KpnS-19N, belong to the the A2 myovirus morphotype formerly belonging to the order *Caudovirales* and *Myoviridae* family (except for Kpn19N belonging to siphovirus B1 morphotype). These phages were investigated further using electron microscopy. The summary of average dimensions of phage particles is provided in Table 4. Electron micrographs of selected phages tested in this study presents Figure 4.

### 3.5. The Influence of Physico-Chemical Factors on Phage Lytic Activity

An important aspect that should be considered in selecting therapeutic phages is their stability. A plethora of factors including the chemical nature of phage particles determines the probability of being unstable in a solution [67]. Thus, we compared the stability of phages from our historical and new collection. 

#### 3.5.1. *Klebsiella* phages from the Przondo-Hessek Collection

The decrease of titer of 2 *Klebsiella* phages by 0.5–1 orders of magnitude at 4 °C and at −70 °C after 3 months (Figure 5) of storage and the decrease of titer of 3 *Klebsiella* phages by 0.5–2 orders of magnitude at 4 °C and at −70 °C after 12 months of storage was observed (Figure 6).

Phages were relatively stable at 4 °C for a period of 24 months (Figure 7). The titer of 1 *Klebsiella* phage decreased by 1 order of magnitude under the influence of chloroform after 24 h. The titer of 4 phages decreased by 1–2 orders of magnitude at a temperature of 60 °C for 10 min. (Figure 8). The highest decrease of phage activity in pH 4 after 5 h (by 0.5–2 orders of magnitude) was observed for 4 phages (Table 5).

#### 3.5.2. Newly Isolated *Klebsiella* Phages

The decrease of titer of 2–3 *Klebsiella* phages by 0.5–3 orders of magnitude at 4 °C and at −70 °C after 3 months (Figure 5) of storage and the decrease of titer of 5 *Klebsiella* phages by 0.5–3.5 orders of magnitude at 4 °C after 12 months (Figure 6) of storage and by 0.5–6 orders of magnitude at −70 °C after 12 months of storage was observed. The titer of 6 *Klebsiella* phages decreased by 0.5–2 orders of magnitude under the influence of chloroform after 24 h. The titer of 6 phages decreased by 1–4 orders of magnitude at a temperature of 60 °C for 10 min. (Figure 8). The highest decrease of phage activity in pH 4 after 5 h for 5 phages by 0.5–4 orders of magnitude was observed. The summarized data is presented in Figure 5, Figure 6, Figure 7 and Figure 8 and Table 5.

In general, the titers of 4 examined *Klebsiella* phages from the Przondo-Hessek collection were more stable than titers of 6 newly isolated *Klebsiella* phages at 4 °C and at −70 °C after 3 (Figure 5), 12 (Figure 6) and 24 months (Figure 7) of storage and in chloroform, at 60 °C (Figure 8) and in pH 4 (Table 5). The obtained results suggest that phages isolated in the 1960s are more stable in terms of lytic activity under different environmental conditions than newly isolated phages. It is possible that several rounds of storage and rejuvenation of historical phages caused selection of their derivatives adapted to longer storage conditions.

### 3.6. General Characteristic of Kpn35, Rpl1, Kpn5N and Kpn6N Genomic Sequences

We sequenced the genomes of bacteriophages from four phage preparations: two phage preparations from the Przondo-Hessek historical collection (*Raoultella planticola* 1/Rpl1—previously named Kpn2/K59 and Kpn35/J24552) and two newly isolated phages (Kpn5N/K737 and Kpn6N/K737). Surprisingly, one of those preparations (Kpn35/J24552) contained DNA fragments whose sequences assembled into one complete genome (designated here as Kpn35c1) and one partial genome with one gap (designated here as Kpn35c2) (Table 6).

DNA sequence reads of all phages, but 35c2 assembled into circular contigs, with no well separated regions of significantly increased coverage, which is indicative of terminally redundant and cyclically permuted virion DNA molecules characteristic for phages that pack their DNA by a headful mechanism (Table 6). The predicted small and large terminase subunits of all four phages appeared to be similar to the small and large terminase subunits of phage T4 or *Aeromonas* phage 44RR2.8t at the amino acid sequence or structural level (Appendix A). Phage T4 and *Aeromonas* phage 44RR2.8t pack their DNA to virion heads by the headful mechanism [68]. Similarities between terminase subunits have been used as a predictor of DNA packaging strategies [69]. Thus, we conclude that all four phages pack their DNA by a headful mechanism and arbitrarily set the genomic regions encoding small terminase subunits at the beginning of each sequence, although it causes the sequence files to open just before the stop codon of tail sheath stabilizer and completion protein gene, which overlaps with the small terminase subunit gene in all these phages (Figure 9 and Figure 10).

**Table 6 pharmaceutics-15-00434-t006:** Genome Features of Phages from Selected Preparations. * Phages have been classified according to the new recommendations of the International Committee for Virus Taxonomy that are based of phylogenetic relations between phages, abolish the order *Caudovirales*, and families *Myoviridae*, *Siphoviridae* and *Podoviridae*, and introduce new phylogeny-based families [70,71]. The assignments of selected genes function are based on results shown in Appendix A.

Phage Preparation	Kpn35/	Kpn35/	Rpl1	Kpn5N	Kpn6N
**Phage name**	Klebsiella phage Kpn35c1	Klebsiella phage Kpn35c2	Raoltella phage Rpl1	Klebsiella phage Kpn5N	Klebsiella phage Kpn6N
**Short name**	Kpn35c1	Kpn35c2	Rpl1	Kpn5	Kpn6
**Morphology**	myovirus	myovirus	myovirus	myovirus	myovirus
**Genus (Species)** *	*Slopekvirus*	*Jiaodavirus*	*Slopekvirus*	*Jiaodavirus*	*Jiaodavirus*
**Family**	*Straboviridae*	*Straboviridae*	*Straboviridae*	*Straboviridae*	*Straboviridae*
**Subfamily**	-	*Tevenvirinae*	-	*Tevenvirinae*	*Tevenvirinae*
**Host(s)**	*Klebsiella pneumoniae*	*Klebsiella pneumoniae*	*Raoltella planticola/Klebsiella pneumoniae*	*Klebsiella pneumoniae*	*Klebsiella pneumoniae*
**Genome size (bp)**	171,960	>167,021	174,270	164,863	163,912
**Virion DNA/packaging strategy**	terminally redundant, circularly permuted/headful packaging,	terminally redundant, circularly permuted/headful packaging,	terminally redundant, circularly permuted/headful packaging,	terminally redundant, circularly permuted/headful packaging, predicted preferred *pac* cuts between pos. 457–480 of genomic sequence	terminally redundant, circularly permuted/headful packaging, predicted preferred *pac* cuts between pos. 457–480 of genomic sequence
**Best Blast hit (coverage in %/identity in %**	Klebsiella phage K15 (98/99.6)	NT	Klebsiella phage K15 (97/98.64)	Klebsiella phage Kpn6 (97/96.2)	Klebsiella phage KP185 (97/95.5)
**G + C content (%)**	41.9	NT	41.9	39.5	39.6
**Number of predicted protein-coding genes**	270	NT	271	271	271
**tRNA genes**	2	NT	2	16	16
**Number of genes encoding homologs of T4 proteins**	102	NT	102	159	159
**Homing endonucleases**	2: gp11, gp134	NT	1: gp134	3: 130, 133, 261	0
**Predicted receptor binding proteins**	1: gp56	NT	1: gp56	1: gp80	1: gp80
**Possible depolymerase domain containing proteins**	3: gp54, gp56, gp262	NT	3: gp54, gp56, gp262	4: gp78, gp79, gp80, gp262	4: gp78, gp79, gp80, gp262

The basic genome features of phages Kpn35c1, Rpl1, Kpn5N and Kpn6N are summarized in Table 6. The sizes of all genomes are within the range between 163 to 172 kb, similar to that of phage T4 (168 bp) [72]. Their GC content (39.5–41.9%) is significantly lower than that of their hosts, *K. pneumoniae* (57.0–57.4%) [73,74] and *R. planticola* (55.6%, as calculated for the reference genome, GenBank acc. no, NZ_CP026047.1). The numbers of predicted proteins encoded by each phage are similar (270–271) (Figure 9; Table 6). Additionally, phages Kpn35c1 and Rpl1 encode two tRNAs each and phages Kpn5N and Kpn6N encode 16 tRNAs each, respectively (Appendix A). 

Phage Kpn35c1 and Rpl1 are highly similar to each other (95.6% identity over the whole genome length) (Appendix A). They are also highly similar to *Slopekvirus* genus phage K15 of the *Starboviridae* family representing the *Slopekvirus* K15 species. This allows us to classify both these phages to the *Slopekvirus* K15 species, according to the phage species demarcation criteria (Figure 9 and Appendix A) [75].

Phages of the *Slopekvirus* genus are represented in the GenBank by 31 complete phage genomes (GenBank acessed on 20 January 2023) that share over 70% of their nucleotide sequence. All these phages, like phages Rpl1 and Kpn35c1, have genomes ranging from about 168 kb to 177 kb. Only a few of them have been described [33,76,77,78] The *Slopekvirus* genus phages are distant relatives of T4 [71]. Our calculations of Kpn35c1 and Rpl1 similarities to T4 at the protein sequence level indicate that 102 of 270 and 272 proteins of each of them, respectively, are similar to proteins of T4 with E < 10^−5^ (Table 6). The similarities include the main core proteins of T4, e.g., main virion structural and morphogenetic proteins, proteins degrading host DNA at infection, replication, recombination, DNA methylation and nucleotide metabolism proteins, host RNA-polymerase modifying and regulating proteins, DNA packaging proteins as well as cell lysis and lysis inhibition proteins. 

The genomic sequences of Kpn35c1 (171,960 bp) and Rpl1 (174,270 bp) are mosaics of genomic sequences of known *Slopekvirus* genus phages, with the exception of a 320-bp region of the Kpn35c1 genome (coord. 102,916–103,235), which is 92% identical to the genome region of the *Citrobacter* phage Moroon of *Tevenvirinae* subfamily of *Straboviridae* (MH823906.1) encoding two hypothetical proteins. The genome of Rpl1 is 2310 bp larger than that of Kpn35c1. Nine regions of the Rpl1 sequence, that together encode 12 proteins and parts of two other proteins (including two putative homing endonucleases) do not have counterparts in the genome of Kpn35c1 (Figure 9). Only two regions of the Kpn35c1 genome (encoding a hypothetical protein and a putative homing endonuclease) do not have counterparts in the genome of Rpl1 (Figure 9). 

Orthologous proteins of Rpl1 and Kpn35c1 are in most cases identical or nearly identical. Only some of them differ significantly in their amino acid sequence or its parts. This includes the C-terminal region of small terminase subunits of both phages, which in Rpl1 has a centrally located 15-aa deletion compared to its Kpn35c1 counterpart. The region corresponding to Rpl1 gene 228, which encodes the predicted peptide-modifying radical SAM enzyme, in the genome of Kpn35c1 is split into two genes potentially encoding two domains of this enzyme. The region corresponding to gene 229, which encodes the putative HNH family homing endonuclease in the genome of Rpl1, in the genome of Kpn35c1 contains three small oppositely oriented genes.

Genomic sequences of Kpn5N and Kpn6N are 95.1% identical, and they are both 93% identical to the genomic sequences of *Klebsiella* phages Kpn6 and Kpn185 representing the *Jiaodavirus* genus (Table 6, Appendix A). Based on the species and genus demarcation criteria, one can conclude that phage Kpn5N and Kpn6N represent a new species of this genus, which we propose to designate *Jiaodavirus 5N* (Appendix A). The *Jiaodavirus* genus phages have been classified to the *Tevenvirinae* subfamily of *Straboviridae*, which includes, e.g., model phage T4 [71]. In T4, DNA packaging from a genome concatamer can start from multiple sites, but the preferred *pac* site is in the 3’ region of gene 16 for the small terminase subunit, within the 24-bp sequence (5’-GAGGCTCAAGAAGCTCGTGAGAAG) [79,80]. Of the specific 24-bp sequence in the small terminase subunit gene of T4, 20 out of 24 bp are conserved in the relevant regions of Kpn5N and Kpn6N (genome coord. 473–496 in both phages) confirming a similar mechanism of DNA packaging and suggesting the involvement of similar sequences as the preferred packaging initiation sites (Table 6 and Appendix A). 

Over half of Kpn5N and Kpn6N proteins (159 out of 271) have orthologs in T4 (10^−5^) (Table 6, Figure 10). The similarities include nearly all core proteins: main virion structural and morphogenetic proteins, proteins degrading host DNA at infection, replication, recombination and nucleotide metabolism proteins, host RNA-polymerase modifying and regulating proteins, DNA packaging proteins and cell lysis as well as lysis inhibition proteins. All head and tail components of Kpn5N and Kpn6N have T4-encoded counterparts, with the exception of the distal part of the long tail fiber, which is over 300 amino acid residues longer and is similar to the L-shaped tail fibers of phage T5. The number of DNA and RNA metabolism- and host RNA polymerase regulation-associated genes of Kpn5N and Kpn6N that are homologous to those of T4 is larger than that in phages Kpn35c1 and Rpl1.

Genomic sequences of Kpn5N and Kpn6N are mosaics of the genomic sequences of 35 other *Jiaodavirus* genus phages deposited in GenBank (accessed on 20 January 2023), with just one exception—a short region between two tRNA-encoding genes. Four protein-coding genome regions of Kpn5N are absent from Kpn6N (Figure 6). They encode a protein of unknown function (gp52) and three homing endonucleases (gp130, gp133 and gp261). All of the 5N homing endonuclease-encoding genes are upstream of the predicted essential phage genes encoding dCTPase, DNA primase and phage baseplate wedge initiator, respectively. The first and the last of them are, respectively, similar in sequence and genomic localization to homing endonucleases found in different *Jiaodavirus* phages: MobE of phage vB_KpM-Wobble (CAD5241756; 99% identity) which is common in the genomes of T-even phages [81], and a putative homing endonuclease of phage vB_KpnM_KpV477 (YP_009288841; 98% identity). The second homing endonuclease of Kpn5N is similar to the GIY-YIG family homing endonuclease of *Salmonella* phage STP4-a (YP_009126235.2; 63% identity) suggesting that its gene was acquired by phage Kpn5N or its ancestor recently in evolution via horizontal gene transfer. Compared to phage Kpn6N, phage Kpn5N lacks five protein-coding genome regions. They encode a part of the capsid and scaffold protein (Kpn6N gp17), and four hypothetical proteins of unknown function (Kpn6N gp186, gp187, gp240 and gp243). The vast majority of Kpn5N proteins have counterparts of identical or nearly identical sequence among the proteins of Kpn6N. The least similar proteins of these two phages include gp176 of Kpn5N which shares only 52% of the amino acid sequence with its counterpart encoded by Kpn6N (gp173), L-shaped tail fibers (74% identity, gp80 of Kpn5N and Kpn6N) and the tail fiber assembly catalysts (65% identity, gp81 of Kpn5N and Kpn6N). Interestingly, the L-shaped tail fibers of Kpn5N and Kpn6N not only differ significantly from each other but are only 92% identical to their most closely related proteins: the L-shaped tail fiber of *Klebsiella* phage Kpn6 (QEG11584.1), and the L-shaped tail fiber of *Klebsiella* phage vB_KpnM_KpV477 (YP_009288928), respectively, suggesting that host specificities of Kpn5N and Kpn6N among *Klebsiella* strains are unique. The tail fiber assembly catalysts are distant homologs of T4 gp38, which catalyzes the organization of long tail fiber proteins [82,83].

Gp17s of phage Kpn5N and Kpn6N are similar in sequence and predicted structure to bacteriophage T4 highly immunogenic capsid decoration protein Hoc, but they differ in size (Appendix A). The difference results from a deletion in the central region of Kpn5N gene 17 as compared to gene 17 of Kpn6N. Such deletion is uncommon in *Jiaodavirus* genus phages as detected by a Blastp search (acessed 3 June 2022). Gp17 of Kpn5N (373 aa) entirely resembles T4 Hoc (NP_049793.1; 376 aa), while Gp17 of Kpn6N is larger (468 aa). Its best matching region of structural similarity to the entire T4 Hoc is between residues 106–468. However, the remaining N-terminal part of Kpn6N gp*17* is also structurally similar to the T4 Hoc, namely to the Hoc N-terminal part, suggesting that Kpn6N gp17 evolved via duplication of the N-terminal fragment of the Hoc-like protein in an ancestral phage. Three proximal domains of four-domain T4 Hoc have an immunoglobulin-like fold [84,85] implying that while Kpn5N gp17 has three immunoglobulin-like domains, Kpn6N gp17 has four immunoglobulin-like domains, like Hoc-proteins of other T4 related bacteriophages [86]. The T4 Hoc protein was shown to markedly induce the production of IgM and IgG antibodies [87]. Thus, the lack of one Ig-like domain in Kpn5N gp17 might potentially result in the lower immunogenicity of Kpn5N compared to other *Jiaodavirus* phages.

### 3.7. Evaluation of the Coding Potential of New Phages with Respect to Their Safety in Therapeutic Applications

#### 3.7.1. Lifestyle and Probability to Participate in DNA Transfer between Bacteria

Phages considered as safe in therapeutic applications should be unable to lysogenize their hosts and unable to transfer genetic material between bacteria by transduction [88]. Additionally, they should not encode any toxins and virulence or antimicrobial resistance determinants. Similarities of core genome genes of Rpl1, Kpn35c1, Kpn5N and Kpn6N to those of T4 and the lack of homologs of lysogeny-associated genes indicate that all these phages are obligatorily lytic, as is T4. Although it contrasts with the formation of turbid plaques by all of them, a possible explanation of this discrepancy could be the emergence of resistance to these phages in bacteria. This is consistent with the results of Townsend et al. [33] who observed that bacteria sensitive to certain *Slopekvirus* or *Jiaodavirus* genus phages can regrow in liquid cultures upon lysis of most cells caused by infection with these phages.

The predicted packaging strategy of Rpl1, Kpn35c1, Kpn5N and Kpn6N DNA to phage heads by a headful mechanism is associated with the ability to pack host DNA, which poses a risk of it transferring to newly infected cells by transduction. However, phages, which like T4, degrade host DNA at infection and use its degradation products as building blocks to synthesize its own DNA are generally considered to be non-transducing [89]. Remnants of host DNA are present in infected cells only in trace amounts during phage DNA packaging, which practically eliminates the production of transducing particles. Host DNA degradation in T4-infected cells occurs through cutting DNA into large pieces by a pair of phage enzymes: rare nicking endonuclease II initiating degradation of cytosine-containing DNA and cytosine-specific endonuclease IV encoded by *denA* and *denB* gene, respectively [90]. Further degradation to mononucleotides is catalyzed by phage gene *46*- and *47*-encoded exonuclease [72,91,92,93,94]. Phages Kpn5N and Kpn6N encode proteins highly similar to all these T4 enzymes, while phages Kpn35c1 and Rpl1 encode homologs of T4 endonuclease II as well as T4 gp*46* and gp*47* (Appendix A) indicating that all these phages degrade the DNA of infected cells.

#### 3.7.2. Potential to Encode Toxins or Other Virulence-Associated Proteins

A comparative analysis of Rpl1, Kpn35c1, Kpn5N and Kpn6N proteins with proteins in the databases did not reveal any significant similarities to known toxins, and virulence or antibiotic-resistance determinants at the protein sequence level. However, predicted structures of certain protein domains of these phages show some similarities to the structures of certain domains of virulence associated proteins. For instance, a major part of phage Rpl1 gp230 has structural similarities to the relevant part of *Shigella* invasin IpaB of identical length (5WKQ_A, probability 72.4% as calculated by HHPred) and the N-terminal part of Rpl1 gp183 has structural similarities to domain A of the human von Willebrand factor (1Q0P_A, probability 99.09% as calculated by HHPred). In *Pseudomonas aeruginosa*, a protein with similarities to domain A of the von Willebrand factor was shown to be surface exposed and to induce *Pseudomonas* virulence [95]. The structural similarities of the mentioned Rpl1 phage proteins to virulence-associated proteins may indicate a common evolutionary origin. Whether the Rpl1 protein domains similar in structure to the domains of virulence factors may play a role in the virulence of phage hosts or interact with eukaryotic cells or proteins remains to be verified. Most likely, they evolved to play phage functions that are not associated with bacterial virulence, as e.g., a homolog of staphylococcal virulence associated autolysin SceD encoded by *Kayvirus* genus phages, which functions as a phage endolysin [96].

### 3.8. Evaluation of the Coding Potential of New Phages with Respect to the Efficacy in Infecting Various Klebsiella Strains

The utility of therapeutic phages depends on their ability to infect a wide range of strains of a given pathogenic species. Bacteria evolved several mechanisms protecting them from phage infection. They can prevent phage adsorption or entry to cells, degrade infecting phage nucleic acid or cause the infected cell to commit a suicide thus blocking out the infection spread [97]. The host range of a given phage depends on the ability to overcome the action of these mechanisms. 

#### 3.8.1. Ability to Penetrate through External Anti-Phage Cell Barriers

A main barrier of *Klebsiella* cells to phage entry is their polysaccharide capsule. Despite the ability of *Slopekvirus* and *Jiadavirus* genus phages to productively infect or lyse from without *Klebsiella pneumoniae* strains of various capsular types, no polysaccharide depolymerases have been identified in these phages [33]. Likewise, our search of predicted Kpn35c1, Rpl1, Kpn5N and Kpn6N gene products for homologs of polysaccharide depolymerases at the protein sequence level revealed no results. This does not conform to the properties of most other described *Klebsiella* phages whose receptor binding proteins typically contain domains of polysaccharide depolymerase activity and their number correlates with phage host range [98,99]. For instance, 8 putative depolymerases were identified among predicted gene products of the *Klebsiella* Jumbo phage infecting strains of various capsular types [66]. A hallmark of phage depolymerase activity is the formation of plaques surrounded with a semi-transparent zone (the so-called halo zone). The haloes result from overproduction of free, enzymatically active receptor binding proteins (tail fibers or tail spikes) in addition to complete viruses [100,101]. Phages Kpn35c1 and Rpl1, as well as Kpn5N and Kpn6N formed plaques with a halo zone on cell layers of certain sensitive strains (Figure 3). Moreover, each of them appears to encode tail fiber or baseplate proteins that contained a domain of predicted structure similar to that of substrate-binding or catalytic domains of certain exopolysaccharide degrading enzymes (Appendix A, Figure 9 and Figure 10). Additionally, while no structural homologs of the major central parts of predicted L-shaped tail fiber proteins of Rpl1 and Kpn35c1 (gp56 in both phages) as well as Kpn5N and Kpn6N (gp080 in both phages) could be identified, the predicted structure of the C-terminal part of all these proteins is similar to the structure of C-terminal parts of the L-shaped tail fiber of bacteriophage T5, the intramolecular chaperone of endo-N-acetylneuraminidase of Enterobacteria phage K1F tail spike, and the C-terminal parts of certain other virion components (Appendix A). All these regions of structural similarities represent the intramolecular chaperone domains of their cognate proteins. They participate in the formation of a triple-β-helix fold characteristic of these proteins, are removed by autocleavage from the remaining parts of the proteins upon folding completion, and, in the case of tail fiber or tail spike proteins, uncover the receptor binding domains or exopolysaccharide depolymerase domains [102,103]. Most likely, Rpl1 and Kpn35c1 gp056, as well as Kpn5N and Kpn6N gp080 function as receptor binding proteins. Whether the proteins of *Jiaodavirus* and *Slopekvirus* phages with structural similarities to depolimerase domains have depolymerase activity remains to be tested.

#### 3.8.2. Ability to Overcome Intracellular Phage Defense Mechanisms

A main barrier that protects bacteria from phage infection by the degradation of incoming phage DNA is formed by restriction-modification systems. T4-related phages overcome the action of these systems by various modifications of bases in their DNA (summarized in Nikulin and Zimin [104] and references therein). Our search of phages Kpn35c1 and Rpl1 predicted gene products for homologs of DNA base modification-associated proteins of T4-even related phages, listed in Nikulin and Zimin [104] revealed only homologs of two T4 proteins involved in DNA modification, namely dCTPase/dUTPase, and DenA (Appendix A). In phage T4 dCTPase/dUTPase hydrolyzes dCTP and dCDP, preventing its incorporation into phage DNA in place of deoxy-hydroxymethyl-cytosine, which is used by T4 as a substrate for DNA synthesis. DenA is a site-specific endonuclease II (EndoII) which initiates degradation of unmodified cytosine-containing DNA [105]. Predicted proteomes of phages Kpn5N and Kpn6N appeared to contain homologs of dCMP hydroxymethylase, which in T4 produces deoxy-hydroxymethyl-cytosine, and α-glucosyl transferase, which in T4 adds glucose in alpha linkage to deoxy-hydroxymethyl-cytosine residues in T4 DNA (Appendix A). However, in place of T4 homolog of β-glucosyl transferase, they encode a homolog of β-1,6-glucosyl-α-glucose transferase—an enzyme that was identified previously in certain T-even related phages, as responsible for β-glucosylation of α-glucosylated ^hm^dCMP in DNA (reviewed in Petrov et al. [106]). Similar homologs of DNA modifyinfg enzymes were identified previously among predicted proteins of phages representing different species of *Jiaodavirus* than phages analyzed here [105]. Taken together, our data indicate that all four phages analyzed by us modify their DNA. While the kind of DNA modification by phages Kpn35c1 and Rpl1 cannot be predicted without further studies, phages Kpn5N and Kpn6N, by prediction, modify their DNA by incorporating deoxy-hydroxymethyl-cytosine in place of deoxycytosine, and by glycosylation of DNA at deoxy-hydroxymethyl-cytosines. Additionally, phages Kpn5N and Kpn6N make further use of cytosine modification in their development. They encode a homolog of the Alc protein, which in T4 acts as an inhibitor of host transcription by attenuating elongation of transcription on cytosine-containing DNA and blocks transcription of the host and other phages [107,108].

Some phages can protect themselves from certain anti-phage defense mechanisms that cause abortive infection. All four phages analysed here appear to encode homologs of T4 gp*rIIA* and gp*rIIB* proteins protecting T4 infected cells from premature lysis caused by prophage lambda-encoded RexA and RexB proteins, and by P2-related resident prophages (Appendix A) (reviewed in [72,109,110]). Apparently, Kpn35c1, Rpl1, Kpn5N and Kpn6N can rely on similar mechanisms of protection from bacterial or phage encoded RexA and RexB-based anti-phage defense mechanisms to that of T4. Our analysis of *Klebsiella pneumoniae* genomic sequences deposited in Genbank revealed that over 130 of them encode homologs of RexA and RexB (data not shown). Presumably, the *Slopekvirus* and *Jiaodavirus* gp*rIIA* and gp*rIIB* homologs contribute to the wide host-range of these phages by protecting them from the action of *Klebsiella* RexA and RexB proteins.

The *Slopekvirus* and *Jiaodavirus* phages analysed here could infect between 20–28% and 19–29% strains of our *Klebsiella* collections tested, respectively. This is consistent with the results obtained by other groups in studies on phages representing these genera [33,76,77,78,111]. Additionally, one jiaodavirus was isolated as infecting *Salmonella* strain (GenBank acc. no. ON550260.1), while some slopekviruses were isolated as infecting strains of *Enterobacter* or *Escherichia* species, indicating that the host range of at least some of these phages exceeds the strains of *Klebsiella* genus. Also, phage Rpl1 described here can productively infect a strain of *R. planticola* in addition to several *Klebsiella* strains. We could identify only a few genomic features that may contribute to the ability of slopekviruses and jiaodaviruses to infect numerous *Klebsiella* strains as well as some strains of other enterobacterial genera. However, taking into account that functions of over half of genes of these phages, especially those that do not have orthologs in T4, cannot be predicted, further studies are likely to uncover a broader repertoire of such features. 

Significant similarities of *Slopekvirus* and *Jiaodavirus* phages to the model phage T4 of obligatorily lytic lifestyle make these phages particularly useful in studies on their therapeutic applications. Moreover, our results and the results of other authors indicate a high antibacterial potential of these phages in targeting their host strains. For instance, one of the described slopekviruses, *Escherichia coli* phage phT4A, was shown to be effective with a nearly 6 order of magnitude reduction of *E. coli* counts after 8 h of incubation with an *E. coli* cell culture [78]. A problem may be the development of phage resistance to slopekviruses and jiaodaviruses suggested by turbid plaques they formed in our studies and by the regrowth of infected bacteria in culture [33]. However, it does not preclude a therapeutic use of these phages in cocktails with phages of other taxons, especially with some of our historical *Klebsiella* phages that have already been used in phage therapy in humans [27,28,29,30,35]. What is more, Henrici De Angelis et al. [112] revealed that acquisition of phage resistance in KPC-producing *K. pneumoniae* strains was associated with decreased virulence.

## 4. Conclusions

Due to the emergence of widespread multidrug-resistant *Klebsiella* strains, there is a growing interest in seeking new strategies to combat antibiotic resistance mainly in health care facilities but also in the food industry and livestock production [113]. In this work we attempted to characterize *Klebsiella* phages from our historical collection, as well as newly isolated phages that are potential candidates for phage therapy and, as mentioned above, some of them have already been used for therapeutic purposes in humans. 

Together, we analyzed 101 bacteriophages including new isolated from various environmental samples and city sewage as well as phages from our old collection being used and stored for decades. Their infectivity was tested against over 700 *Klebsiella* strains and some other enterobacterial strains of clinically relevant species. The abundance of phages infecting strains of various *Klebsiella* species varied in different samples, perhaps as a result of differences in the biodiversity of bacteria in various environmental niches. Phages infecting *K. pneumoniae* or *K. oxytoca* were the most numerous among our isolates. They represented the myovirus, podovirus and siphovirus morphotypes of tailed phages. While most of the phages had rather narrow host-range, some isolates could infect over 50% of the tested strains, which is unexpected given the high variability of *Klebsiella* exopolysaccharide capsules. Surprisingly, no substantial differences could be observed between the host-range of new collection and old collection phages. However, phages from the old collection outperformed new isolates in their stability under conditions of various pH, temperature and in the presence of chloroform.

Host range is a key factor when selecting phages suitable for therapy [114]. One of the advantages of a narrow host range is selective killing without disturbing non-pathogenic bacteria inhabiting the body. On the other hand, it constitutes an obstacle in rapid selection of therapeutic phages when time can be precious in life-threating cases. In fact, a recent case report published by Qin et al. [115] focuses on difficulties in phage selection and administration in patients with complex UTI caused by a total of 21 *K. pneumoniae* strains. Thus, our results justify the organization of large collections for the long-term deposition and storage of phage isolates to target any possible pathogenic bacterial strain.

Based on DNA homologies, the analyzed phages could be classified to two genera of the *Straboviridae* family, which includes bacteriophage T4 and several phages previously known as T4-like [71]. Similarities between predicted products of their genes orthologous to those of T4 revealed their obligatorily lytic developmental strategy and certain mechanisms that may contribute to the wide host range of all four phages, implying the advantage of choice for therapeutic applications phages for which well explored models exist.

Herridge et al. [3] listed over 100 described and classified *Klebsiella* phages with a rather promising conclusion on their therapeutic usefulness. The growing number of papers focusing on *Klebsiella* phages, their isolation, biology, genetic and serology characteristics clearly indicates that this topic will be intensively investigating in the future. This will be developed in parallel with rising obstacles in phage treatment, which may soon be difficult to overcome, and the exploration of phages’ non-antibacterial activity [116,117,118].

## Figures and Tables

**Figure 1 pharmaceutics-15-00434-f001:**
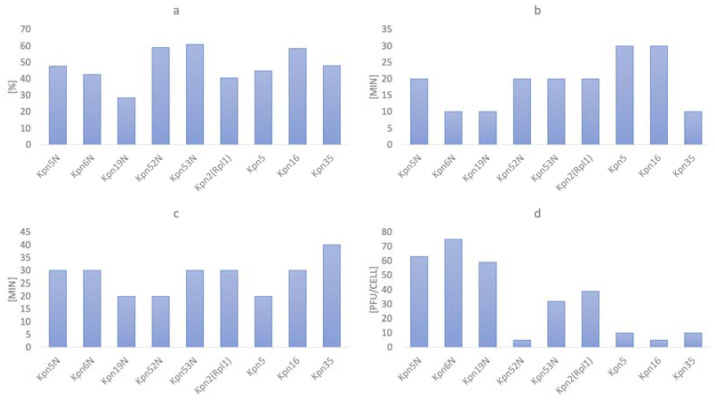
One-step growth results for 9 tested phages: (**a**) Adsorption rate; (**b**) Adsorption time; (**c**) Latent period; (**d**) Burst size. All experiments were performed in triplicate and the provided results constitute the mean value of each set of repetitions.

**Figure 2 pharmaceutics-15-00434-f002:**
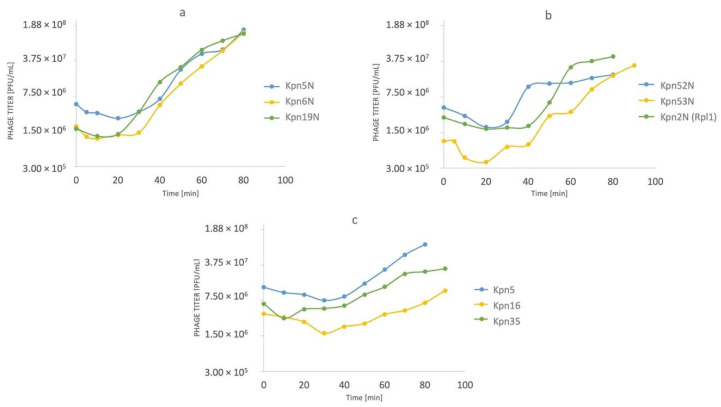
Phage infection kinetics for 9 tested phages: (**a**) phages Kpn5N, Kpn6N, Kpn19N; (**b**) phages Kpn52N, Kpn53N, Kpn2N (Rpl1); (**c**) phages Kpn5, Kpn16, Kpn35. All experiments were performed in triplicate and the provided results constitute the mean value of each set of repetitions.

**Figure 3 pharmaceutics-15-00434-f003:**
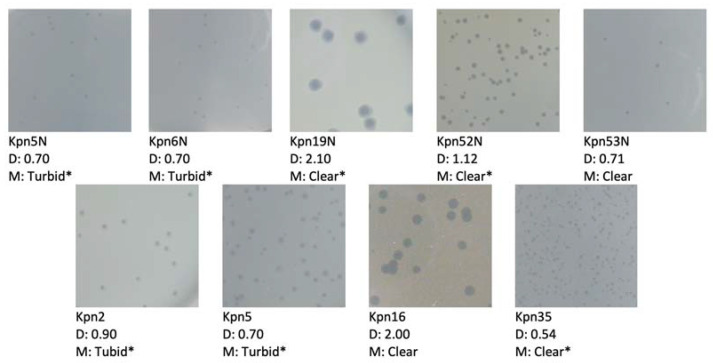
Plaque morphology of 9 tested phages photographed on agar plates. D—plaque diameter [mm]; M—plaque morphology; * presence of halo.

**Figure 4 pharmaceutics-15-00434-f004:**
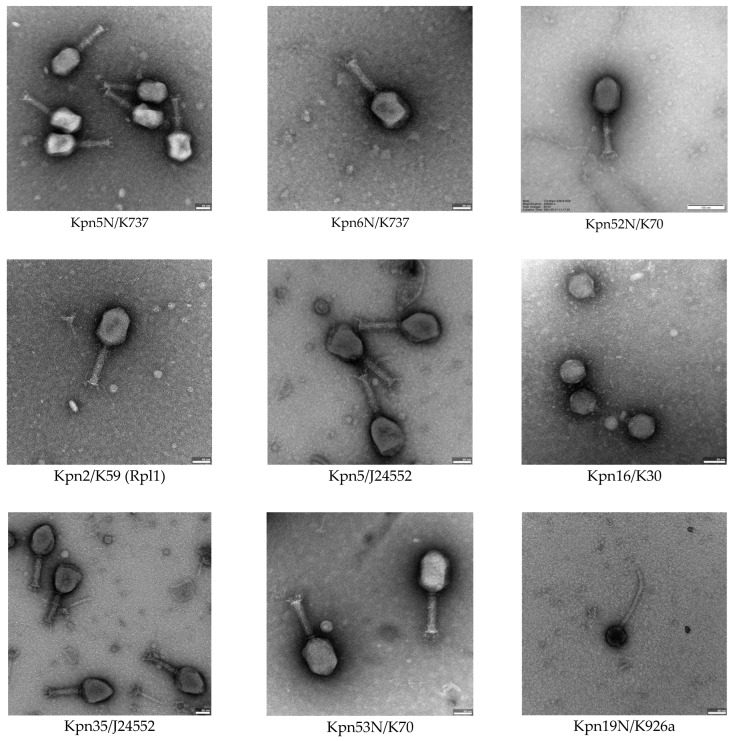
Electron micrographs of selected phages tested in this study. TEM magnification: 80,000–250,000.

**Figure 5 pharmaceutics-15-00434-f005:**
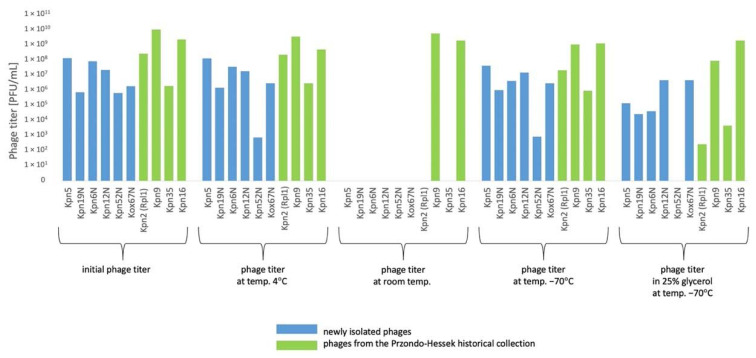
Phage lytic activity at different storage conditions after 3 months.

**Figure 6 pharmaceutics-15-00434-f006:**
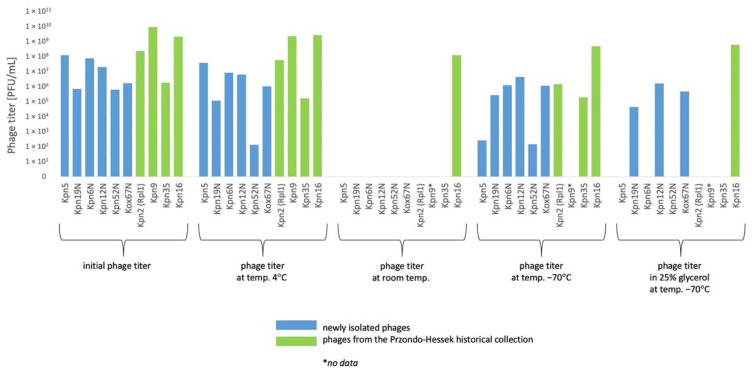
The influence of various storage conditions on the phage lytic activity after 12 months.

**Figure 7 pharmaceutics-15-00434-f007:**
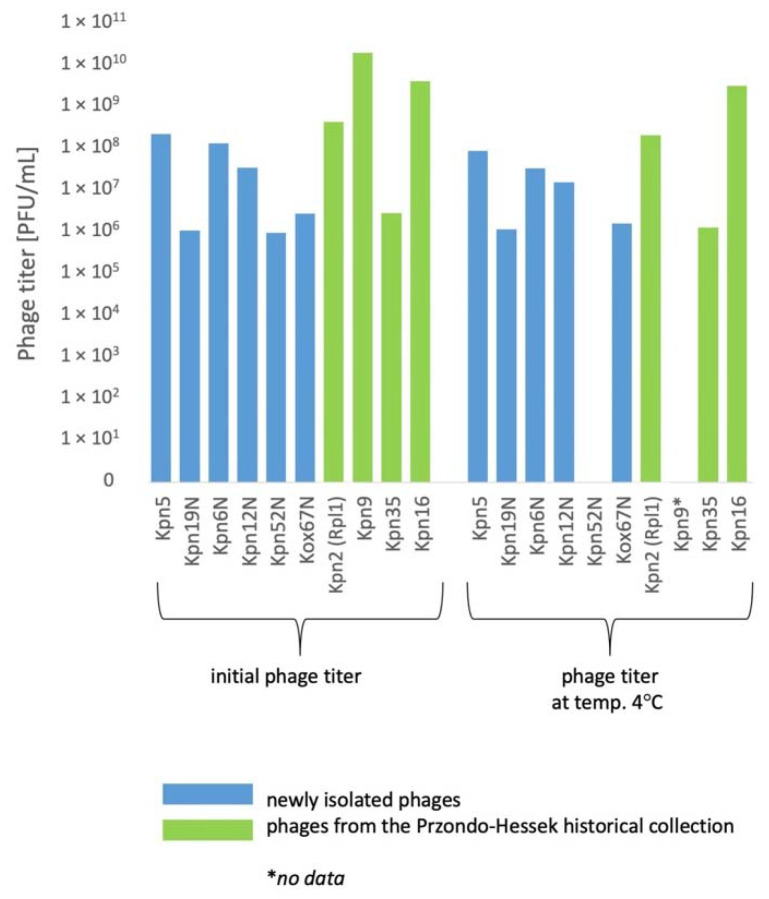
The influence of various storage conditions on the phage lytic activity after 24 months.

**Figure 8 pharmaceutics-15-00434-f008:**
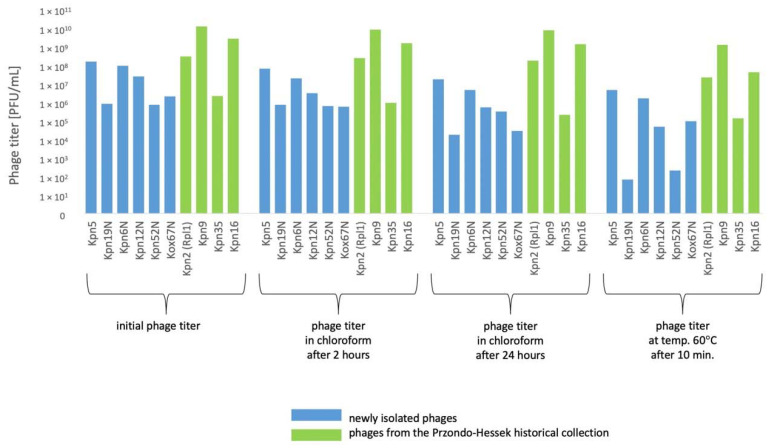
The influence of chloroform and 60 °C temperature on the phage lytic activity.

**Figure 9 pharmaceutics-15-00434-f009:**
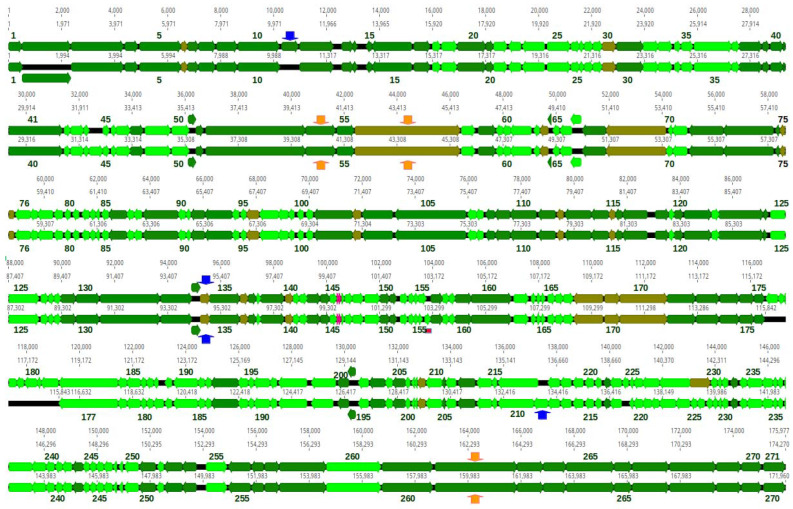
Alignment of the genomic maps of bacteriophages Rpl1 (upper row) and Kpn35c1 (lower row). Protein coding genes are in different shades of green. Dark green indicates genes that have T4 phage counterparts encoding similar proteins, light green represents genes that do not. Olive green represents genes encoding proteins with similarity of only certain domains to domains of T4 proteins. Red arrowheads represent tRNA encoding genes. Orange vertical arrows indicate genes whose products contain regions of predicted structural similarity to certain regions of polysaccharide degradindg enzymes. Dark blue vertial arrows indicate genes encoding homing endonucleases. The region of Kpn35c1 genome, which does not have a counterpart in the genomes of other *Slopekvirus* genus phages is indicated by the red rectangle underneath the Kpn35c1 genome map. The region of Kpn35c1 genome which does not have a counterpart in the genome of Rpl1 (coord. 133,549–134,088; 539 bp) encodes a putative homing endonuclease. The regions of Rpl1 genome that do not have counterparts in the genome of Kpn35c1 are at coord: 348–523 (175 bp, central region of small terminase subunit), 10,198–10,942 (744 bp, homing endonuclease), 103,013–103,112 (100 bp, hypothetical protein), 115,729–118,376 (2647 bp, 7 hypothetical proteins), 122,147–122,347 (200 bp, hypothetical protein), 136,275–136,334 (59 bp, non-coding region), 139,143–139,403 (260 bp, hypothetical protein), 141,217–142,151 (934 bp, N-terminal part of radical SAM containing protein, GIY-YIG domain containing protein [putative homing endonuclease]) and 142,391–142,532 (141 bp, hypothetical protein). Every fifth protein coding gene of Rpl1 and Kpn35c1 genome is numbered in bold above and below the Rpl1 and Kpn35c genome map, respectively. Numbers above each sequence in the alignment show the relevant genome coordinates. Numbers above each row of the alignment show the alignment coordinates including gaps.

**Figure 10 pharmaceutics-15-00434-f010:**
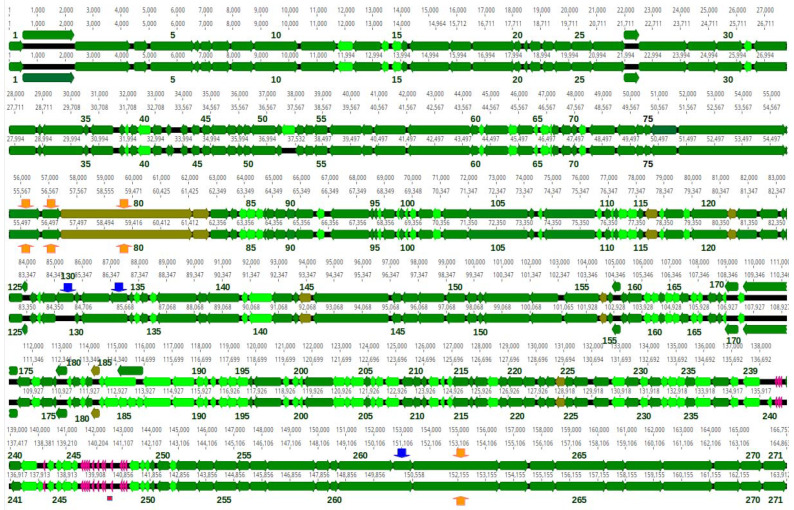
Alignment of the genomic maps of bacteriophages Kpn5N (upper row) and Kpn6N (lower row). Protein coding genes are in different shades of green. Dark green indicates genes that have T4 phage counterparts encoding similar proteins, light green indicates genes that do not. Olive green represents genes encoding proteins with similarity of only certain domains to domains of T4 proteins. Red arrowheads represent tRNA encoding genes. Orange vertical arrows indicate genes whose products contain regions of predicted structure similar to polysaccharide degradindg enzymes. Dark blue vertial arrows indicate genes encoding homing endonucleases. The region of Kpn6N genome between the genes encoding tRNA-Mer-CAT and tRNA-Asp-GTC (coord. 140,338–140,441) that does not have a counterpart in the genomes of other Jiaodavirus genus is indicated by the red rectangle underneath the alignment. Every fifth protein coding gene of Kpn5N and Kpn6N genome is numbered in bold above and below the Kpn5N and Kpn6N genome map, respectively. Numbers above each sequence in the alignment show the relevant genome coordinates. Numbers above each row of the alignment show the alignment coordinates including gaps.

**Table 1 pharmaceutics-15-00434-t001:** Bacterial strains from our collection used for phage host range determination.

Bacterial Strain	Number of Strains	Strain Origin
*Klebsiella pneumoniae* PCM 1 (ATCC 13886 )	1	PCM 1 (ATCC 13886)
*Klebsiella pneumoniae*	14	Environmental sample
*Klebsiella pneumoniae*	262	Clinical strain
*Klebsiella oxytoca*	144	Clinical strain
*Klebsiella oxytoca*	15	Environmental sample
*Klebsiella variicola*	2	Clinical strain
*Klebsiella ozenae*	9	Clinical strain
*Raoultella planticola*	2	Clinical strain
*Klebsiella aerogenes*	2	Clinical strain
*Pantoea agglomerans*	2	Clinical strain
*Pluralibacter gergoviae*	5	Clinical strain
*Enterobacter cowanii*	3	Clinical strain
*Escherichia coli B*	1	HIIET collection

**Table 2 pharmaceutics-15-00434-t002:** Origin of isolated *K. pneumoniae* and *K. oxytoca* phages.

Place of Sample Collection	Sample Type	Year of Isolation	Isolated Phages	
			*Klebsiella pneumoniae*	*Klebsiella oxytoca*
Wastewater treatment plant in Opole	Incubated city sewage	2008	6N, 7N, 8N, 10N	9N, 13N
		2009	19N, 33N, 38N, 47N	18N, 42N
	Crude concentrated city sewage	2009	37N	15N
		2010	21N, 48N, 61N	16N, 49N
University Teaching Hospital in Opole	Incubated hospital sewage	2010	52N, 60N	63N
		2011	65N	-
		2013	-	67N
		2015	-	B40
4th Military Hospital in Wrocław	Incubated hospital sewage	2010	53N	-
Wastewater treatment plant in Kraków-Płaszów	Incubated city sewage	2008	5N	2N, 3N
		2009	-	41N, 46N
	Crude concentrated city sewage	2009	11N, 20N, 34N, 36N, 39N	14N, 17N
		2010	55N	-
Wastewater treatment plant in Pleszew (Greater Poland province)	Crude city sewage	2010	12N, 35N, 50N, 54N, 62N	56N, 64N
Wastewater treatment plant in Grodzisk Mazowiecki (Mazovian Province)	Crude water sample	2012	-	68N
		2015	-	WI42, WI43, WI105, WI106
	Incubated water sample	2015	-	WII43
Wrocław, irrigated pool	Crude sewage from irrigated pool No. 1	2008	4N	-
Chojnów (Lower Silesia province), water intake	Crude water sample	2009	32N	-
Czeczotki (Mazovian Province), water intake	Crude water sample	2015	-	WI53
Vistula river (Lesser Poland Province)	Crude water sample	2009	15N, 27N, 28N, 29N, 30N, 41N *	42N, 43N, 44N, 45N *

* Following MALDI-TOF mass spectrometry system for bacterial strain identification, 6 phages with symbols 27N, 28N, 29N, 30N, 44N, 45N turned out to be *Raoultella terrigena* phages and 5 phages named 15N, 16N, 41N, 42N, 43N were classified to *Enterobacter hormaechei* phages.

**Table 3 pharmaceutics-15-00434-t003:** The broadest phage lytic spectra among tested phages.

HIIET Phage Collection			Przondo-Hessek Phage Collection		
Phage Symbol	ICTV Symbol	Number and Percentage of Sensitive Strains	Phage Symbol	ICTV Symbol	Number and Percentage of Sensitive Strains
Kpn53N	vB_KpnM-53N	73 (35.2%)	Kpn2 *	vB_RapM-1	58 (28.0%)
Kpn5N	vB_KpnM-5N	59 (28.5%)	Kpn16	vB_KpnM-16	52 (25.1%)
Kpn52N	vB_KpnM-52N	54 (26.0%)	Kpn35	vB_KpnM-35	43 (20.8%)
Kpn6N	vB_KpnM-6N	39 (18.9%)	Kpn5	vB_KpnM-5	31 (14.9%)
Kpn19N	vB_KpnS-19N	33 (16.0%)	Kpn9	vB_KpnM-9	17 (8.2%)

* Identification performed with MALDI-TOF mass spectrometry system resulted in classifying the original host strain K59 of phage *K. pneumoniae* 2 (Kpn2) to *R. planticola* strain (Rpl1).

**Table 4 pharmaceutics-15-00434-t004:** Virion dimensions of selected phages tested in this study as determined based on electron micrographs.

Phage Symbol/ Host Symbol	ICTV Phage Symbol	Morphotype	Total Lenght	Dimension of Head [nm]		Dimension of Tail [nm]		
				Capsid Length	Capsid Width/Diagonal	Total Length	Tail Width	Base Plate Width
Kpn5N/K737	vB_KpnM-5N	A2	230.4	115.1	81.6/106.4	115.3	20.0	31.3
Kpn52N/K70	vB_KpnM-52N	A2	221.1	107.6	82.2/98.6	118.5	22.7	37.2
Kpn53N/K70	vB_KpnM-53N	A2	229.8	118.3	85.5/100.1	111.5	20.1	32.8
Kpn6N/K737	vB_KpnM-6N	A2	237.6	119.4	90.2/109.5	118.2	21.3	32.9
Kpn19N/K926a	vB_KpnS-19N	B1	234.8	56.1	53.6/178.7	178.7	9.1	-
Kpn2/K59 (Rpl1)	vB_RaM-1	A2	231.7	113.9	83.6/117.8	117.8	20.6	32.3
Kpn16/K30	vB_KpnP-16 *	C1	-	-	-	-	-	-
Kpn35/J24552	vB_KpnM-35	A2	234.8	111.1	88.5/123.7	127.7	21.7	35.3
Kpn5/J24552	vB_KpnM-5	A2	224.7	108.4	88.8/100.1	116.3	20.7	31.7

* results pending.

**Table 5 pharmaceutics-15-00434-t005:** The influence of pH on the phage lytic activity.

No.	Phage/Host	Initial Phage Titer PFU/mL	Phage Titer at pH 4 PFU/mL		Phage Titer at pH 5 PFU/mL	
			After 1 h	After 5 h	After 1 h	After 5 h
1	Kpn5N/K737 ^	1.5 × 10^8^	9.5 × 10^7^	3.6 × 10^7^	9.5 × 10^7^	8.2 × 10^7^
2	Kpn19N/K917 ^	8.0 × 10^5^	1.1 × 10^5^	0	8.2 × 10^5^	8.7 × 10^3^
3	Kpn6N/K737 ^	3.0 × 10^6^	2.6 × 10^6^	1.6 × 10^6^	3.0 × 10^6^	3.6 × 10^6^
4	Kpn12N/K1195 ^	4.9 × 10^3^	8.7 × 10^2^	0	2.2 × 10^3^	6.2 × 10^2^
5	Kpn52N/K1920 ^	7.2 × 10^5^	2.6 × 10^5^	2.8 × 10^3^	6.7 × 10^5^	5.2 × 10^5^
6	Kox67N/K3459 ^	5.0 × 10^4^	1.0 × 10^4^	0	3.4 × 10^4^	7.0 × 10^3^
7	Kpn2/K59 (Rpl1) *	1.7 × 10^8^	1.1 × 10^8^	6.5 × 10^7^	1.8 × 10^8^	1.6 × 10^8^
8	Kpn9/J24552 *	1.2 × 10^10^	2.9 × 10^9^	1.4 × 10^9^	2.9 × 10^9^	1.6 × 10^9^
9	Kpn35/J24552 *	2.1 × 10^6^	3.0 × 10^5^	1.7 × 10^5^	3.7 × 10^5^	2.0 × 10^5^
10	Kpn16/K30 *	2.0 × 10^9^	1.6 × 10^9^	1.4 × 10^7^	1.7 × 10^9^	4.7 × 10^8^
**No.**	**Phage/host**	**Initial phage titer** **PFU/mL**	**Phage titer ** **at pH 6 ** **PFU/mL**		**Phage titer** **at pH 8 ** **PFU/mL**	
			**After ** **1 h**	**After** **5 h**	**After** **1 h**	**After ** **5 h**
1	Kpn5N/K737 ^	1.5 × 10^8^	1.6 × 10^8^	1.1 × 10^8^	1.0 × 10^8^	1.0 × 10^8^
2	Kpn19N/K917 ^	8.0 × 10^5^	8.2 × 10^5^	1.7 × 10^4^	3.1 × 10^5^	1.7 × 10^4^
3	Kpn6N/K737 ^	3.0 × 10^6^	4.0 × 10^6^	3.4 × 10^6^	3.0 × 10^6^	2.3 × 10^6^
4	Kpn12N/K1195 ^	4.9 × 10^3^	5.0 × 10^3^	2.8 × 10^3^	3.5 × 10^3^	3.0 × 10^3^
5	Kpn52N/K1920^	7.2 × 10^5^	6.0 × 10^5^	7.0 × 10^5^	7.4 × 10^5^	6.8 × 10^5^
6	Kox67N/K3459 ^	5.0 × 10^4^	3.7 × 10^4^	2.1 × 10^4^	3.0 × 10^4^	1.9 × 10^4^
7	Kpn2/K59 (Rpl1) *	1.7 × 10^8^	1.8 × 10^8^	1.7 × 10^8^	1.4 × 10^8^	1.5 × 10^8^
8	Kpn9/J24552 *	1.2 × 10^10^	2.9 × 10^9^	2.6 × 10^9^	2.9 × 10^9^	2.6 × 10^9^
9	Kpn35/J24552 *	2.1 × 10^6^	5.7 × 10^5^	3.7 × 10^5^	2.7 × 10^5^	3.2 × 10^5^
10	Kpn16/K30 *	2.0 × 10^9^	2.1 × 10^9^	1.8 × 10^9^	2.2 × 10^9^	2.0 × 10^9^

^ newly isolated phages; * phages from the Przondo-Hessek historical collection.

## Data Availability

The presented data are available on request from the corresponding author.

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
