# Peer review of "Characteristics of Environmental Klebsiella pneumoniae and Klebsiella oxytoca Bacteriophages and Their Therapeutic Applications"

_pharmaceutics, 2023, doi:10.3390/pharmaceutics15020434_

Round 1
Author Response
We thank the reviewer for all comments and suggestion that helped us to improve our manuscript substantially.
Our responses to the comments are below.
The abstract is too long and should be more concise.
We have modified the abstract as recommended.
Table 1 and 2 could in the supplementary; The legends in Figures 1 and 2 could have larger fonts and the grid eliminated; Table 4 Figure 3 Table 5 Figure 4 could be in one figure like this one.
We have modified several tables/figures including grid elimination, increasing font size and merging some tables/figures into one. However, due to the amount of data we kept the table with electron microscopy measurements as a separate item.
Table 6-9 could be transformed into graphs that are more readable
We have transformed these tables into graphs to increase readability.
The description of the genomic sequences is long and sometimes mixes results and discussion.
To shortening the results and discussion, and to avoid repetitions of results in the Discussion section we followed the suggestion of reviewer 3 and merged the Results and Discussion sections. This also allowed us to follow the recommendations of Reviewer 2 and 3 and add Conclusions as the last part of our manuscript.
Table 11 could be more detailed and summarize at least partly the results.
We thank the reviewer for this helpful suggestion. Former Table 11 was reorganized and supplemented with additional data to help readers to follow the description of phage genomic features presented in the Results and Discussion section. It is Table 10 in the revised version of our manuscript.
It would be interesting to have the genomic maps and genomic comparison(s) in a figure in the text rather than in the supplementary. There are many softwares for the presentation of phage genomes and their comparison such as Phamerator for example.
We transferred the genomic maps with comparisons between the genomes of our Slopekvirus and Jiaodavirus isolates to the main text of the manuscript as requested. We attempted to use other software to produce these maps, but because the sequence differences between related phages of each pair are below 5%, our current figures with manually edited comparisons to point out significant differences seemed to be more informative and thus we decided to leave the maps in their original form.
The similarity with other viruses may be shown in a phylogenetic tree. Evaluation of the Coding Potential of New Phages with Respect to their Suitability for Therapy In this paragraph, again there is a mix of results and discussion. In my opinion, the two paragraphs should be reorganized in order to separate the results and the discussion which should therefore also be restructured.
We restructured the manuscript in its parts containing the results and discussion and merged the Result and Discussion sections by moving certain parts of the discussion into the relevant positions of the former Results section. This allowed us to eliminate many repetitions in the discussion and helped to improve the manuscript. The major findings of the manuscript are summarized shortly in the form of conclusions.
As mentioned by the authors:” The utility of therapeutic phages depends on their ability to infect a wide range of strains of a given pathogenic species”.
This aspect is not developed enough. For example, some recently described Jumbo phages infecting Klebsiella pneumoniae are even not mentioned (Genomic characterization of four novel bacteriophages infecting the clinical pathogen Klebsiella pneumoniae.
Bonilla E, Costa AR, van den Berg DF, van Rossum T, Hagedoorn S, Walinga H, Xiao M, Song W, Haas PJ, Nobrega FL, Brouns SJJ. DNA Res. 2021 Aug 25;28(4):dsab013. doi: 10.1093/dnares/dsab013.). Interestingly, the vBKpM_FBKp24 (φKp24) jumbo myophage encodes at least nine tail fibers containing different depolymerase domains, which suggests an expanded host range.
We thank the reviewer for paying our attention to this manuscript. We cited it according to the reviewer's recommendations but also used this well organized manuscript as a guide to reorganize our manuscript by merging the Results and Discussion section, and by adding conclusions. Additionally, to emphasize the genomic feature of our phages that may contribute to their relatively wide host-range we reorganized the last section of results by separating the description of features that may be associated with different mechanisms of host-range expansion, such as the potential to encode exopolysaccharide depolymerases, modify phage DNA to avoid restriction or overcome bacterial abortive infection systems.
Reviewer 2 Report
Authors present the results of a long-term study on isolation and biology of bacteriophages active against K. pneumoniae, as well as K. oxytoca strains and those obtained from the Bacteriophage Laboratory (BL) and Phage Therapy Unit (PTU) of the Hirszfeld Institute of Immunology and Experimental Therapy (HIIET) in Wroclaw, Poland, and Medical University of Warsaw, Poland. A total of 101 bacteriophages were isolated from the Vistula River in addition to those from hospital and city sewage collected from municipal wastewater treatment plants across the country using the enrichment and colorimetric methods. This work is meaningful in this work.
1 The language should be polished by native English speaker.
2 Some significant references, 10.1093/bib/bbab462, 10.1016/j.chemolab.2021.104351, should be discussed in this work.
3 It was suggested that authors should show an outlines of this work.
Author Response
We thank the reviewer for helpful suggestions.
Our responses to the comments are below.
The language should be polished by native English speaker.
As recommended the manuscript was verified for the correctness of English (we have attached the certificate of proofreading).
Some significant references, 10.1093/bib/bbab462, 10.1016/j.chemolab.2021.104351, should be discussed in this work.
We thank the reviewer for paying our attention to the mentioned publications. We have read them carefully, but noticed that they concern protein modifications and the identification of disease-related compounds with the use of sophisticated algorithms and databases, which exceeds the scope of our manuscript. Thus, we have not cited these references. However, we will keep in mind the referred manuscripts for the purpose of our future projects.
It was suggested that authors should show an outlines of this work.
We thank the reviewer for this suggestion. We included the outline of our results in the Conclusions section, which have been added as the last part of our manuscript.
Reviewer 3 Report
The manuscript entitled"Characteristics of environmental Klebsiella pneumoniae and Klebsiella oxytoca bacteriophages and their therapeutic applications" highlights the identification and therapeutic use of Klebsiella bacteriophages. The study is well-planned with a comprehensive methodology and also explains various aspects of bacteriophages and their promising usage in therapeutics.
However, I have little concern about the results section. The authors intentionally overexplained the results section. I suggest that the discussion part must be kept separate while explaining results or at least both the Results and Discussion parts can be merged. OR, at least reduce irrelevant or unnecessary material from the result section. As the authors already explained the possible reasons for a phage to become competent or effective phages in the introduction section. So, there is no need to explain again and again in other parts of the manuscript.
Secondly, I suggest adding the conclusion part to this manuscript as it is missing. It is necessary for such kind of study to conclude the outcomes for the greater interest of readers.
Thanks
Author Response
We thank the reviewer for helpful comments and suggestions.
Our responses to the comments are below.
The authors intentionally overexplained the results section. I suggest that the discussion part must be kept separate while explaining results or at least both the Results and Discussion parts can be merged. OR, at least reduce irrelevant or unnecessary material from the result section. As the authors already explained the possible reasons for a phage to become competent or effective phages in the introduction section. So, there is no need to explain again and again in other parts of the manuscript.
We reorganized the manuscript according to the recommendations. To shorten the results and discussion, and to avoid the repetitions of results in the Discussion section we merged the Results and Discussion sections.
I suggest adding the conclusion part to this manuscript as it is missing. It is necessary for such kind of study to conclude the outcomes for the greater interest of readers.
The conclusion part was added at the end of manuscript as suggested to summarize and outline our results.